# Leveraging Deep Learning for Visual Odometry Using Optical Flow

**DOI:** 10.3390/s21041313

**Published:** 2021-02-12

**Authors:** Tejas Pandey, Dexmont Pena, Jonathan Byrne, David Moloney

**Affiliations:** Intel Research & Development, W23 CX68 Leixlip, Ireland; dexmont.pena@intel.com (D.P.); jonathan.byrne@intel.com (J.B.); david.moloney@intel.com (D.M.)

**Keywords:** visual odometry, ego-motion estimation, deep learning

## Abstract

In this paper, we study deep learning approaches for monocular visual odometry (VO). Deep learning solutions have shown to be effective in VO applications, replacing the need for highly engineered steps, such as feature extraction and outlier rejection in a traditional pipeline. We propose a new architecture combining ego-motion estimation and sequence-based learning using deep neural networks. We estimate camera motion from optical flow using Convolutional Neural Networks (CNNs) and model the motion dynamics using Recurrent Neural Networks (RNNs). The network outputs the relative 6-DOF camera poses for a sequence, and implicitly learns the absolute scale without the need for camera intrinsics. The entire trajectory is then integrated without any post-calibration. We evaluate the proposed method on the KITTI dataset and compare it with traditional and other deep learning approaches in the literature.

## 1. Introduction

Visual odometry [1] is the challenging task of camera pose estimation and robot localization based only on visual feedback. It represents one of the fundamental problems of computer vision, and forms an integral part of Simultaneous Localization And Mapping (SLAM) and robot kinematics. It is a long-standing challenge in the field, and is being worked on for several decades [2,3].

Visual odometry forms an integral component for autonomous agents. A self-driving agent is responsible for being aware of the environment and other moving objects in the scene, as well as its own relative movements. Given a sequence of frames, the agent needs to keep track of the relative inter-frame motion, as well as positioning on a global scale. Marginal errors called “drift” are introduced in the relative motion prediction. They accumulate over the length of a path and propagate throughout the global trajectory, leading to significant errors.

Classical visual odometry is dominated by stereoscopic and RGB-D cameras where information such as scale, which gets lost during projection mapping, can be recovered to reduce the drift [4,5,6,7]. Others rely on sensor fusion, where data from multiple sensors, such as LIDAR, IMU, and GPS are fused to come up with more reliable and robust solutions [8,9]. Recently, there has been an increased interest in problems regarding ego-motion with the rise of autonomous agents navigating in unfamiliar dynamic environments, such as self-driving cars and UAV drones, leading to a demand for cheap and reliable commodity solutions. Integrating multiple cameras with different sensors requires not only space and expensive hardware, but also sufficient processing power and energy. Even stereoscopic cameras are constrained if the robot is too small and cameras are too close. Monocular visual odometry has the advantage of using only a single camera; however, it is significantly more challenging due to the number of ambiguities involved. These challenges include loss of scale, dynamic environments, inconsistent changes in illumination, and occlusion [10]. Traditional approaches can be divided into two categories: direct, and feature-based.

*Feature-based* methods [11,12,13] extract geometrically consistent features [14] and track them across subsequent frames. The camera motion is estimated by minimizing re-projection errors between feature pairs. Though feature-based methods are robust to noise, they fail to work in sparse, textureless environments where the algorithm is unable to generate suitable features.

*Direct methods* [15] track the intensity of pixels and estimate camera motion by minimizing photometric errors. They are more robust in environments lacking visual features, but are susceptible to sudden changes between subsequent frames.

In computer vision, deep learning-based approaches have outperformed traditional methods in tasks such as object classification and detection [16,17,18]. However, these approaches are unsuitable for geometric tasks, as they were designed to be translation/orientation invariant, rather than equivariant. They lose track of spatial relationships and ignore physical features such as size, velocity, and orientation. Different adaptions were designed to overcome this problem [19,20].

In this paper, we first study the problem of visual odometry using deep neural networks. We search for deep learning approaches to overcome the limitations present in direct and feature-based methods. Deep neural networks learn feature representations in higher dimensions and can replace the engineering effort required in integrating and fine-tuning individual components [1,21].

We then propose a pipeline using CNNs that takes optical flow as input and directly estimates a relative 6-DOF camera pose. We augment the network with RNNs, as represented in Figure 1, to implicitly model relationships between subsequent frames in higher dimensions.

The rest of the paper goes as follows—Section 2 reviews related work and Section 3 comprises our proposed model, followed by experimental results in Section 4 and the conclusion in Section 5.

## 2. Related Work

Monocular visual odometry is the task of tracking the flow of information from a single camera and predicting the path undertaken by it. Traditional methods [12,22,23] are solely insufficient and have to rely on external parameters to account for the ambiguities involved, such as scale, whereas deep learning approaches [24,25] are self-sufficient and show improved results compared to older traditional methods.

Recent deep learning approaches [26,27] propose taking a pre-trained FlownetS [28] and re-purposing it to predict relative poses. The FlownetS architecture, given two RGB images, uses a single network for feature extraction and matching to estimate optical flow. The hypothesis is that the features learned for capturing the relationship between frames for estimating optical flow should also be suitable for the task of predicting the relative pose between them.

Another approach is to directly use optical flow as input [26,29] and regress the relative pose based on learned priors. The advantage here is that it also allows us to decouple the problem of visual odomtery into feature extraction and motion estimation.

### 2.1. Sequence-Based Modelling

Wang et al. [24] were the first ones to propose a sequence-based model. They argued that CNNs were not sufficient to model the visual odometry pipeline (VO). While CNNs are useful for learning visual features, they cannot capture motion dynamics. They proposed using LSTMs to capture the temporal relationship between frames. However, LSTMs are not suitable for working with high-dimensional data, such as high-resolution images. Thus, the FlownetS architecture is augmented with LSTMs to model the entire VO pipeline. Feature representations extracted from CNNs are passed onto LSTMs, which output pose estimates at every timestep.

Parisotto et al. [27] countered that while LSTMs can model relationships between frames, they cannot influence past outputs, and proposed an attention-based, neural global-bundle adjustment network. The output features from FlownetS are passed onto the global-adjustment network, which uses temporal convolutions to account for drift over long trajectories. Subsequently predicted trajectories can be refined by recursively passing the outputs back to the global-adjustment network.

Jiao et al. [25] proposed an iterative update to [24] using bi-directional LSTMs, and outperformed [24] on the KITTI dataset. Bi-directional LSTMs model relationships for both forward and backward sequences. LSTMs learn to model the conditional probability between poses. A bi-directional LSTM would open up the probable space by learning from future sequences as well.

### 2.2. Optical Flow-Based Visual Odometry

Costante and Ciarfuglia [30] argued that deep learning approaches involved not only learning visual cues, but camera intrinsics as well. They recommended decoupling the learning of visual and motion features for better generalization. Rather than use a single network for feature extraction and motion estimation, they proposed an auto-encoder to capture an embedding of the optical flow and thereby predict the relative pose based off of it. They formed a multi-objective loss function to minimize the error in reconstruction of optical flow and camera pose estimation, thus ensuring the captured embedding was suitable for motion estimation. They were able to outperform traditional monocular approaches [13,31]. However, a multi-objective function can hinder the performance of motion prediction where the network can struggle to balance the loss of optical flow reconstruction and pose estimation. Moreover, the two tasks may not share favourable features, thus reducing the accuracy of both reconstruction and pose prediction [32].

Costante et al. [29] used optical flow as an input to estimate the camera pose. In order to retain the spatial relationship, they proposed splitting the input optical flow into quadrants and passing individual quadrants into a combination of 5 different neural networks—one for every quadrant, and one for the entire optical flow. Different regions add value for specific directions, and thus, splitting the optical flow into quadrants allowed us to preserve such information from shared parameters. Outputs of all networks were concatenated in order to regress the relative translation and rotation parameters. They were able to outperform traditional monocular methods as well. However, their approach required five networks, which is computationally expensive and memory-intensive.

Muller and Savakis [26] proposed using the same architecture as FlownetS while passing 2D optical flow vectors as input, rather than a six-channel concatenated RGB pair. They used the same hyper-parameters due to FlownetS’ success on RGB images and performed pose estimation using a single network, passing optical flow vectors as input. However, a network designed for learning relationships between RGB pairs may not be sufficient for the task of pose estimation from optical flow, as it was unable to outperform [29].

### 2.3. Contributions

Costante et al. [29] argued that CNNs were translation-invariant, and proposed splitting the optical flow into quadrants to retain motion information. However, their approach was computationally expensive. Thus, our proposed solution is more similar to [26], of conventional regression. However, the approach by Muller and Savakis [26] of adopting the same FlownetS architecture for the task of visual odometry, while promising, could not outperform Costante et al.

Deep learning networks, such as Mobilenet [16] and Resnet [17], could outperform traditional approaches for image classification, but not for optical flow estimation. Thus, Flownets [28] came into inception as networks more suitable for the task of optical flow estimation. Now, rather than using Flownets for the task of pose estimation as well, we propose an architecture optimized for estimating relative poses from optical flow.

Since the inception of Flownets, there has been an increased interest in augmentation with optical flow in deep learning. It has shown promising results with its introduction in applications such as frame interpolation for slow-motion videos [33] and super-resolution [34]. However, optical flow inputs have underperformed in the field of visual odometry. Our work shows the benefits of optical flow for monocular visual odometry.

Wang et al. [24] argued CNNs were insufficient by themselves to model the visual odometry pipeline from raw RGB images, and proposed augmenting CNNs with LSTMs, which can implicitly learn to model the temporal relationship between subsequent frames. To exploit this advantage, we also augment our optical flow-based network with LSTMs to reduce the drift errors in a sequence.

## 3. Visual Odometery Using Optical Flow

In our approach, we combine optical flow-based ego-motion estimation with sequence-based modelling. We take in optical flow as input and use CNNs to learn a dense feature description, which is then passed onto bi-LSTMs. Our network models spatio-temporal relationships across a sequence to frame the entire visual odometry pipeline (Figure 1). We show our proposed architecture in Figure 2. However, calculating optical flow [35] has always been a computationally expensive task. Fischer et al. [28] proposed two novel architectures, FlownetS and FlownetC, for estimating the optical flow given two RGB images. While the FlownetS architecture used a single network for feature extraction and matching, FlownetC used a siamese network for feature extraction and a novel correlation layer for feature matching in higher dimensions. The networks were able to deliver competitive results with real-time frame rates. Flownet was followed by Flownet2 [36], which proposed stacking the different architectures together and significantly improved upon optical flow estimates. Since then, there has been an increased interest in leveraging deep learning for predicting optical flow.

For our proposed method, we use LiteFlowNet [37] to predict our optical flow input. LiteFlowNet was designed on the principles of pyramid feature scaling, feature warping, and cost volume. Rather than scaling images, feature pyramids are generated at different scales to make the network more robust to noise and changes in illumination. It then uses feature warping to generate a cost volume representation, and introduces a regularization layer to refine the optical flow estimate. LiteFlowNet is 30 times smaller than Flownet2, and outperformed PWC-Net [38] and Flownet2 on the KITTI flow 2012 benchmark [39]. We show a visualization of the generated optical flow in Figure 3.

### 3.1. Optical Flow

The rise in popularity of deep learning for image classification came due to the availability of large-scale datasets. Ideally, we would want our dataset to represent the real world. However, there is a lack of comprehensive data for geometric tasks, such as visual odometry, which require recording and manually calibrating data from different sensors. To overcome this limitation, we chose to work with optical flow rather than RGB images, as optical flow is an encoded representation of flow of motion. It is scene-invariant and maps the pixel correspondences between subsequent frames. We believe an optical flow-based method would generalize better compared to RGB images.

### 3.2. CNN Architecture

Extracting features from optical flow is suitable for a CNN, as it requires extracting patterns from encoded flow vectors. However, existing off-the-shelf feature extractors, such as Mobilenet and Resnet, were trained on Imagenet, which is of a different domain. Thus, we propose our own network for extracting features from optical flow. Rather than use the FlownetS architecture which was designed for RGB images, we did a parameter search for a network suitable for optical flow inputs, through hyper-parameter tuning [41]. The network consists of seven convolution layers with max pooling applied after every layer, excluding the last one. We apply a rectified linear unit as our activation function. The CNN architecture is described in Table 1. We use a small architecture to avoid the problem of over-fitting, as we are working on a relatively small dataset.

We pass 2D optical flow vectors as input, where every vector acts as a relative pose for that point in space. At this stage, our network has to implicitly estimate the relative pose for all points and accumulate the results to come up with a final estimate for camera motion.

### 3.3. Sequence-Based Modelling

Sequence-based modelling for visual odometry was first proposed by [24]. RNNs have shown remarkable performance on Natural Language Processing (NLP) tasks [42], and have been used extensively for sequence-based learning [43]. RNNs have the ability to learn relationships over time as they can access feature outputs of the previous cell.

Given feature $x_{t}$ at time *t*, RNN is updated at time *t* by:(1)$$\begin{array}{c} h_{t}=\mu \left(W_{xh}x_{t}+W_{hh}h_{t-1}+b_{h}\right)\\ y_{t}=W_{hy}h_{t}+b_{y} \end{array}$$
where $h_{t}$ is the state variable of the hidden layer at time *t*, $y_{t}$ an output variable. $W_{hh}$ and $W_{xh}$ are the weight matrix of the hidden layers and input features, respectively, and $b_{h}$ and $b_{y}$ are the bias vectors of the hidden layers and input features, respectively. μ is a non-linear activation function, generally consisting of tanh and hard sigmoid.

However, RNNs suffer from vanishing gradients if the gradients pass through many timesteps. Thus, to account for the vanishing gradients, LSTMs were introduced with a three-gate system, namely, an input gate, output gate, and a forget gate. The gates determine how to update the current state. LSTMs avoid the problem of vanishing gradients, and can capture relationships over long periods. For our paper, we follow the approach proposed by [25] and use a bi-directional LSTM which consists of two LSTMs.

The equations for the input gate, forget gate, output gate, and input cell state are as follows:(2)$$\begin{array}{c} f_{t}=\sigma_{g}\left(W_{f}x_{t}+U_{f}h_{t-1}+b_{f}\right)\\ i_{t}=\sigma_{g}\left(W_{i}x_{t}+U_{i}h_{t-1}+b_{i}\right)\\ o_{t}=\sigma_{g}\left(W_{o}x_{t}+U_{o}h_{t-1}+b_{o}\right)\\ \tilde{C}=\tanh\left(W_{C}x_{t}+U_{c}h_{t-1}+b_{c}\right) \end{array}$$
where $W_{f},W_{i},W_{o}$, and $W_{c}$ are the weight matrices for the forget gate, input gate, output gate, and input cell state, while $U_{f},U_{i},U_{o}$ and $U_{c}$ are the weight matrices connecting the previous cell output state to the three gates and the input cell state. $b_{f},b_{i},b_{o}$ and $b_{c}$ are the bias vectors for the forget gate, input gate, output gate, and input cell state. $\sigma_{g}$ is the gate activation function, a hard sigmoid function, and tanh is the hyperbolic tangent function. The cell output state $C_{t}$ and the hidden state $h_{t}$ are as follows:(3)$$\begin{array}{c} C_{t}=f_{t}*C_{t-1}+i_{t}*{\tilde{C}}_{t}\\ h_{t}=o_{t}*tanh\left(C_{t}\right). \end{array}$$

Bi-directional LSTMs consists of two LSTMs. Thus, the output is represented as:(4)$$\begin{array}{c} y_{t}=\sigma \left(\vec{h_{t}},\overset{\leftarrow }{h_{t}}\right), \end{array}$$
where σ is the function used to combine the two output sequences and $y_{t}$ is the output variable at time *t*.

Bi-directional LSTMs learns from both forward-sequence $\vec{h}$ from time $t_{1}...t_{n}$, as well as backward-sequence $\overset{\leftarrow }{h}$ from time $t_{n}...t_{1}$.

The last layer of our CNN block passes a dense feature representation to the bi-LSTMs to model the temporal relationships between feature representations in higher dimensions.

We use a pair of bi-LSTMs as proposed by [24] with 128 units, thus a total of 256 units, and a timestep of five frames. The bi-LSTMs are then connected to a fully connected layer which estimates the 6-DOF pose.

### 3.4. Loss Function

We use the Mean Squared Error (MSE) loss to minimize the euclidean distance between the ground-truth pose and prediction:(5)$$\begin{array}{c} \underset{\theta }{\mathrm{argmin}}\frac{1}{N}\sum_{i=1}^{N}\sum_{k=1}^{K}\left|\right|{\hat{p}}_{k}^{\left(i\right)}-p_{k}^{\left(i\right)}{\left|\right|}_{2}^{2}-w||{\hat{q}}_{k}^{\left(i\right)}-q_{k}^{\left(i\right)}{\left|\right|}_{2}^{2} \end{array}$$
where *p* and *q* denote translation and rotation parameters and *N* and *K* the number of samples and dimensions, respectively.

## 4. Results

In this section, we go over our implementation and training details and evaluate our model on the KITTI dataset [40].

### 4.1. Experimental Setup

The network was trained using Tensorflow 2.0 with an Intel i7-6850K, Nvidia Geforce Titan X Maxwell and 32 GBs of RAM.

### 4.2. Dataset

The KITTI dataset [40] consists of 22 sequences. Only sequences 00–10 have their ground truth provided. The rest are saved for testing and only supplied with raw sensor data. The dataset includes stereo images captured at 10 fps, as well as IMU and LIDAR. For our purposes, we only work with the left-most stereo camera. The dataset consists of a car driving in dense residential urban environments and sparse highways. To train our model, we divide the dataset into training and testing data. We use sequences 00, 02, 04, 06, 08, and 10 for training our model and sequences 03, 05, 07, and 09 for validation. We avoid working with sequence 01 since the car is driving at high speeds, resulting in large displacements between frames.

The ground-truth is comprised of 4 × 4 transformation matrices. We calculate the relative poses for each RGB pair and convert them to an euler representation comprised of three translation and rotation parameters each.

### 4.3. Implementation

We use the Adam optimizer with a learning rate of 0.001. Learning rate decay is employed to achieve faster convergence, and the network is trained for 100 epochs. We first train the network to infer relative poses from optical flow, and then augment it with bi-directional LSTMs.

### 4.4. Results

We evaluate our model on sequences 03, 05, 07, and 09 to realize the generalization of our network. The evaluation is based on the KITTI odometry benchmark [39]. Translation and rotation errors are calculated for sub-sequences of lengths ranging from 100 to 800 m every 100 m. Then, the averaged root-mean-squared error is considered. We compare our results vs. VISO2 [13] and MagicVO [25]. VISO2 is an open-source, feature-based solution for visual odometry. It minimizes re-projection errors between sparse feature matches and supports monocular and stereo-based configurations. We show comparisons against both configurations. MagicVO is a deep learning approach that augments FlownetS with bi-directional LSTMs. We chose to compare against MagicVO, as their results outperform other approaches in the literature, and thus they act as a good baseline. Since their implementation is not public, we compare against results published in their paper. We report our results in Table 2. We also draw a quantitative and qualitative comparison against VISO2. We show quantitative comparisons in Figure 4, and qualitative ones in Figure 5.

### 4.5. Analysis

The averaged quantitative results are given in Table 2. We outperformed our monocular baselines (VISO2_M, MagicVO) on unknown environments and came close to the stereo-based approach (VISO_S). The stereo-based configuration uses multiple perspectives for better scene understanding and reduces the error caused by drift, while our approach uses deep learning to generate robust spatio-temporal feature descriptions, significantly outperforming traditional monocular methods.

The dataset consists of dynamic scenes with multiple moving objects, and was captured at 10 fps with the car moving at high speeds. This results in large displacements between frames. It can prove challenging for traditional feature-based approaches to perform feature-matching between distant frames. A neural network might prove to be invariant to such faults, as the network takes many features into consideration.

Interestingly, LSTMs reduced our accuracy in a counter-intuitive manner. While the LSTMs managed to model the relationships between RGB image pairs, they fail to do so when working with optical flow images. This could be due to the prevalence of temporal feature representations, since RGB image pairs use the same image twice when iterating through a sequence, which is missing when working with optical flow since optical flow is scene-invariant. Subsequent optical flows can differ, whereas the RGB frame at time *t* will be shared with both the frames at time t−1 and t+1. Figure 4b,d indicates that using LSTMs helped at high speeds when there are large displacements between frames, and it is able to use sequential information to output an improved estimate. There are also limited samples in the dataset with the car driving at speeds greater than 50 or slower than 20. We believe more experimentation is required in assessing the role of LSTMs when working with optical flow feature representations.

Figure 5 also indicates the errors are not evenly distributed. While straight roads show little divergence, errors around turns and corners are high, which skews the global trajectory when a turn is encountered. Figure 4b,d shows errors increase significantly when the car is moving at a slow speed. Most of the samples consist of the car driving in a straight manner with uniform speed. The car occasionally slows down to turn or come to a stop. The lack of balanced data is a big drawback for deep learning against a stereo baseline.

With errors concentrated around turns, some frames contribute more to the relative errors which get accumulated over a trajectory. A loss function that can explicitly enforce the relationship between subsequent frames and weigh the contribution of each frame would be ideal. The averaged root-mean-square error is not sufficient where data are not weighted uniformly. Conventional methods perform feature extraction and evaluate whether a frame is viable for pose estimation and can contribute to the global trajectory. However, a neural network, when given an input, will always provide an output. Neural networks assume all inputs have an equal contribution. A biased network that can weight the input can get better results.

## 5. Conclusions

In this paper, we presented a deep learning-based solution for visual odometry (VO). We validated that CNNs are capable of predicting camera motion with only optical flow as input. Our proposed solution outperformed MagicVO, which used features extracted from RGB images and modelled the VO pipeline using bi-directional LSTMs.

While our initial experiments surpassed VISO2_M and MagicVO (monocular baselines), the stereo-based approach still scored better. Additionally, augmenting our approach with LSTMs showed interesting behaviour that we would like to pursue in further studies. However, we believe that a bigger dataset would help in achieving better results.

For future work, we will test and validate our network on more datasets. We will also explore the architecture search space for a network with an induction bias more tuned for visual odometry.

## Figures and Tables

**Figure 1 sensors-21-01313-f001:**
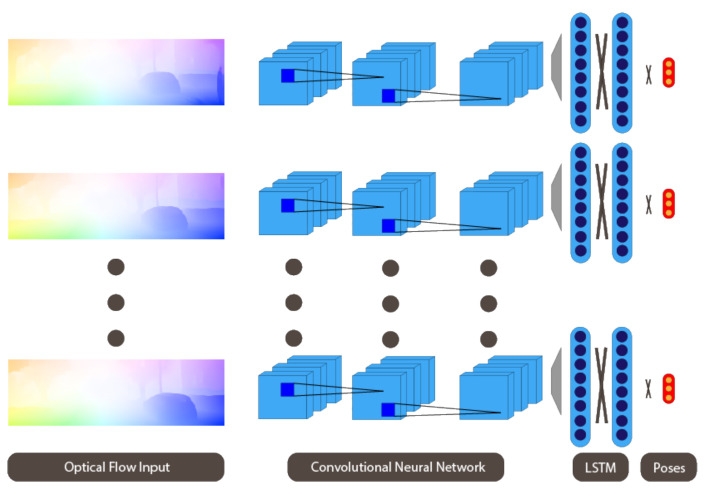
Overview of our approach: Optical flow is passed on to CNNs, extracting relevant features, which are in turn passed on to the RNNs.

**Figure 2 sensors-21-01313-f002:**
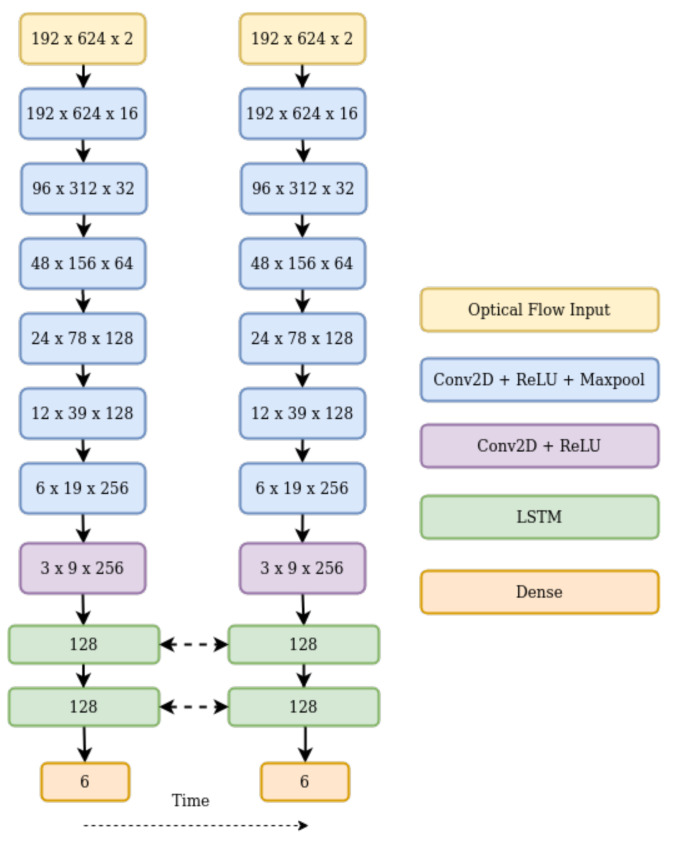
Proposed architecture.

**Figure 3 sensors-21-01313-f003:**
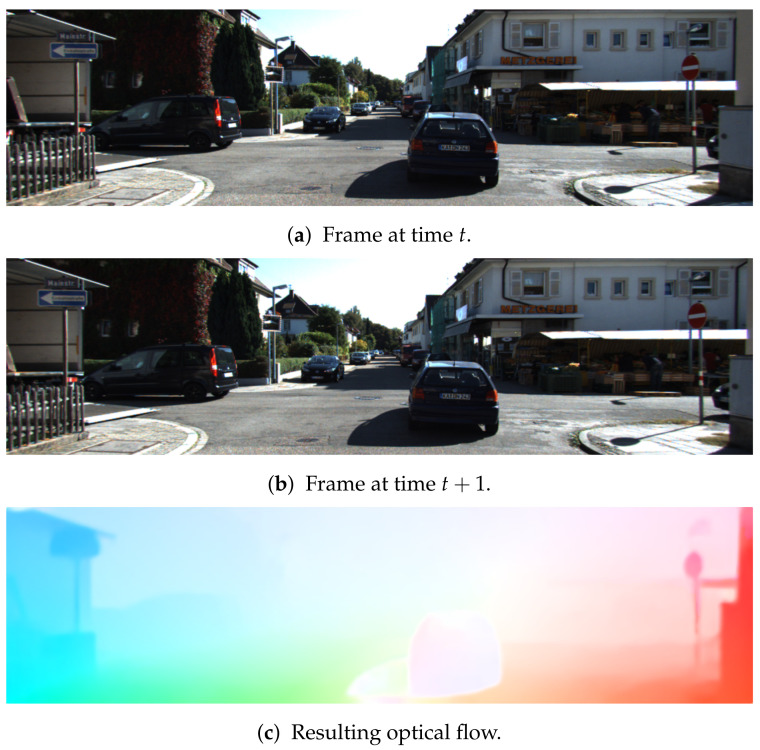
Example frames from the KITTI dataset [40] and resulting optical flow using LiteFlowNet [37].

**Figure 4 sensors-21-01313-f004:**
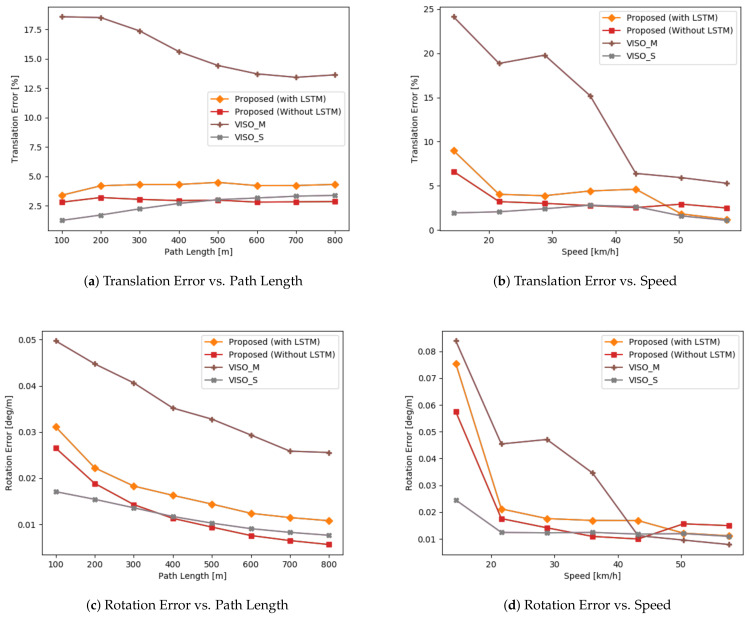
Comparison of the proposed method based on the KITTI evaluation kit.

**Figure 5 sensors-21-01313-f005:**
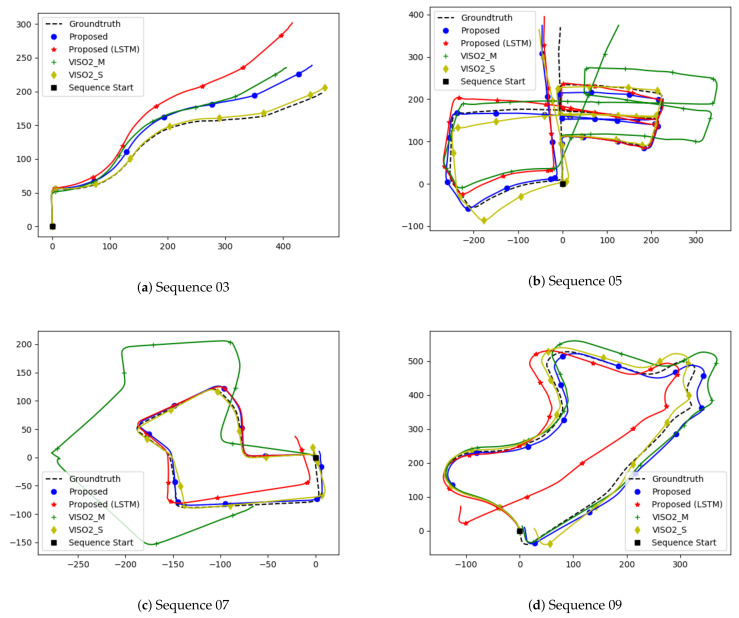
Comparison of our proposed method with VISO2.

**Table 1 sensors-21-01313-t001:** Structure of CNN.

Layer	Kernel Size	Strides	Channels
Conv2D	3	1	16
MaxPool		2	
Conv2D	3	1	32
MaxPool		2	
Conv2D	3	1	64
MaxPool		2	
Conv2D	3	1	128
MaxPool		2	
Conv2D	3	1	128
MaxPool		2	
Conv2D	3	1	256
MaxPool		2	
Conv2D	3	1	256

**Table 2 sensors-21-01313-t002:** Comparison of proposed method vs. monocular and stereo-based VISO2 and MagicVO. • t: average translation drift (%) on length of 100–800 m. • r: average rotation drift (°/100 m) on length of 100–800 m.

Seq.	Proposed		Proposed (LSTM)		VISO2_M		VISO2_S		MagicVO	
	t (%)	r (°)	t (%)	r (°)	t (%)	r (°)	t (%)	r (°)	t (%)	r (°)
03	4.85	2.54	9.82	3.64	10.57	1.73	**2.94**	**1.09**	4.95	2.44
05	2.89	1.22	3.03	1.23	19.02	4.21	2.40	**1.15**	**1.63**	2.25
07	**2.56**	2.15	6.43	3.39	34.16	9.98	2.67	1.61	2.61	**1.08**
09	**2.54**	**0.90**	5.05	1.90	5.76	1.05	2.86	1.14	5.43	2.27
**Avg.**	3.21	1.70	6.08	2.54	17.38	4.24	**2.72**	**1.25**	3.66	2.01

## Data Availability

The KITTI Dataset [40] used for this study can be accessed at http://www.cvlibs.net/datasets/kitti/ (accessed on 27 January 2021).

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
