# Peer review of "Leveraging Deep Learning for Visual Odometry Using Optical Flow"

_sensors, 2021, doi:10.3390/s21041313_

Round 1

Reviewer 1 Report

Interesting study that proposes the joint and direct exploitation of deep learning and optical flow for monocular visual odometry. I believe the paper has potential for publication. However, some aspects must be clarified/completed/revised:

1) Section II should include a subsection entitled (for instance) "Contributions", where the contributions of this study claimed by its authors are explicitly stated with respect to the brief summary of the state of the art previously provided in this section.

2) Please, include in Section 4.1 the training time required for the proposed network. Also, it would be interesting to include some comments about the advantages and/or disadvantages of considering optical-flow images instead of RGB when it comes to training, maybe also in relation with the time required for the hyper-parameter tuning mentioned in Section 3.2.

3) Why is MagicVO excluded from Figs. 4 and 5? The comparison with other approaches (Table 2) is not very broad, so expected information should not be removed from those figures.

4) It'd be nice to provide some performance figures for the embedded realization of the proposed approach. At the end of the day, you'd move to a monolocular solution, most likely trading off accuracy, to save power and form factor. I guess working directly with the optical flow must help reduce the processing workload versus RGB images, but it's that, a guess.

Reviewer 2 Report

This paper presents a deep leaning approach for monocular visual odometry providing vehicle ego-motion from optical flow information. The optical flow vectors derived from subsequent frames are calculated using LiteFlowNet. The latter provide the input of a proposed 7 layers CNN architecture outputting the 6 DOF (translations and rotations) relative camera pose. The inter-relative pose relationship is learned using a RNN providing then the overall trajectory. Finally, in order to prevent possible concerns regarding vanishing gradient in a RNN sequence, a bidirectional LSTM, which comprises forward and backward sequences was implemented. This approach was tested on KITTI dataset and compared against classical monocular and stereo visual (VISO) and deep learning monocular (MagicVO) odometry approaches. The overall structure of this manuscript is well structured and easy to read. Even though the presented work uses a similar methodology of ref 25 (MagicVO) , it differentiates itself by using optical flow vectors instead of RGB images as input, using a more performant deep optical flow estimation approach (LiteFlowNet vs FlowNet), less CNN layers (7 vs 9) and significantly less nodes in the LSTM (256 vs 2000). The presented results shows an overall better performance than MagicVO and a much better performance than monocular VISO. That being said, I have few interrogations concerning the results and analysis sections, which are list below: - First, regarding the use of the dataset. You mentioned in Section 4.2 “We avoid working with sequence 01 since the car is driving at high speeds resulting in large displacements between frames.“. This argument is not a good justification in my opinion. This sequence could have been added to the test sequences and possibly show limitations or in the contrary that your approach is coping very well. However, unless it is tested this cannot be certified. Actually, this argument somehow contradicts what you have shown in figure 4.b and 4.d where respective translation and rotation errors are relatively low at high speed (over 45 km/h). It is in fact at a low speed (under 20km/h - turns, braking, slowing down) that rotational and translational errors are the highest regardless of the chosen approach. - In this regard, by taking the result of MagicVO in sequence 1 (1.12 t% and 1.01 r°) to the others sequences it gives it an overall average of 3.15 t% and 1.81 r°. This does not implies that your method would not necessarily keep providing better overall performances than MagicVO. However, I feel that testing your method on sequence 1 would bring a better overall picture and show full exploitation of the given dataset. - Deep learning approaches are especially appreciated for their potential generalisation. If your method demonstrated convincing results on KITTI, then it would have been extremely interesting to see how this new architecture behaves on other datasets. This is the main reproach I have, especially since your approach requires only images and poses and there are several datasets providing these and which are freely available (CityScape for instance). - You may argue that using another dataset would prevent you to make comparison with MagicVO, but this is not important. If you show that you approach performs similarly well quantitatively (i.e. reproducing similar figures in t% and r°) in other conditions against a ground truth, then you demonstrate likeliness of generalisation which is the selling point of deep learning approaches. - Another point I wish to highlight, is the use of the bidirectional LSTM. You made a mention that your proposed method with LSTM was “counterintuitively” performing worst than your approach with LSTM. If you attributed the potential cause to the fact that optical flow vectors are used as input instead of RGB images pair, I find this is quite shallow in term of justification and this could have been expanded more actually. First, the fact that this may be the only or main cause is not guarantee. Indeed, if the cited DeepVO and MagicVO approaches may be using as well RGB images as input of their CNN, they also differ with your approaches regarding the number of nodes used in there LSTM (1000 and 2000 respectively). This has not been mentioned and this may also be one of the factor affecting your approach using LSTM. Thank you again for your contribution which I appreciated reading. On the overall, I believe you work is relevant and demonstrate interesting points but it also highlights few issues which would be great to be addressed.

Round 2

Reviewer 1 Report

The authors have satisfactorily addressed my comments. I understand that embedded realization was not the focus of this paper. Still, I really hope the authors further research on that aspect of this work because great potential is anticipated.